# Urinary Continence Recovery after Robotic Radical Prostatectomy without Anterior or Posterior Reconstruction: Experience from a Tertiary Referral Center

**DOI:** 10.3390/jcm12041358

**Published:** 2023-02-08

**Authors:** Francesco Sessa, Rossella Nicoletti, Alessio Pecoraro, Paolo Polverino, Anna Rivetti, Francesco Lupo Conte, Mattia Lo Re, Mario Belmonte, Andrea Alberti, Edoardo Dibilio, Maria Lucia Gallo, Alekseja Manera, Mauro Gacci, Arcangelo Sebastianelli, Graziano Vignolini, Sergio Serni, Riccardo Campi, Vincenzo Li Marzi

**Affiliations:** 1Unit of Urological Robotic Surgery and Renal Transplantation, Careggi Hospital, University of Florence, 50100 Florence, Italy; 2Department of Experimental and Clinical Medicine, University of Florence, 50100 Florence, Italy

**Keywords:** continence, post-prostatectomy incontinence, RALP, radical prostatectomy, prostate cancer

## Abstract

Background: The aim of our study is to evaluate the prevalence and predictive factors of short- (30 d) and mid-term continence in a contemporary cohort of patients treated with robotic-assisted laparoscopic prostatectomy (RALP) without any posterior or anterior reconstruction at our referral academic center. Methods: Data from patients undergoing RALP between January 2017 and March 2021 were prospectively collected. RALP was performed by three highly experienced surgeons following the principles of the Montsouris technique, with a bladder-neck-sparing intent and maximal preservation of the membranous urethra (if oncologically safe) without any anterior/posterior reconstruction. (Self-assessed urinary incontinence (UI) was defined as the need of one or more pads per die (excluding the need for a safety pad/die. Univariable and multivariable logistic regression analysis was used to assess the independent predictors of early incontinence among routinely collected patient- and tumor-related variables). Results: A total of 925 patients were included; of these, 353 underwent RALP (38.2%) without nerve-sparing intent. The median patient age and BMI were 68 years (IQR 63–72) and 26 (IQR 24.0–28.0), respectively. Overall, 159 patients (17.2%) reported early (30 d) incontinence. In multivariable analysis adjusting for patient- and tumor-related features, a non-nerve-sparing procedure (OR: 1.57 [95% CI: 1.03–2.59], *p* = 0.035) was independently associated with the risk of urinary incontinence in the short-term period, while the absence of cardiovascular diseases before surgery (OR: 0.46 [95% CI: 0.320.67], *p* ≤ 0.01) was a protective factor for this outcome. At a median follow-up of 17 months (IQR 10–24), 94.5% of patients reported to be continent. Conclusions: In experienced hands, most patients fully recover urinary continence after RALP at mid-term follow-up. On the contrary, the proportion of patients who reported early incontinence in our series was modest but not negligible. The implementation of surgical techniques advocating anterior and/or posterior fascial reconstruction might improve the early continence rate in candidates for RALP.

## 1. Introduction

Robotic-assisted laparoscopic prostatectomy (RALP) has been shown to achieve excellent oncological outcomes with a low rate of complications in patients with prostate cancer [1,2,3,4]. Despite the surgical procedure nowadays being well consolidated [5,6,7], it may have a severe impact on patient quality of life, especially regarding the risk of stress urinary incontinence (SUI) [8,9]. In the literature, the rate of urinary continence recovery at 12 months after surgery ranges from 69% to 96% [3], with a large heterogeneity in definitions of continence and its evaluation (validated questionnaires, number of pads per day, pad test, etc.) [10]. To date, there is still no consensus on the definition of male incontinence after radical prostatectomy or on the objective criteria that should be considered when assessing its severity. This missing data on the grade of post-prostatectomy male incontinence explain the difficulties in identifying what is the best tailored treatment for each incontinent patient [11].

However, several factors have been related to the risk of post-surgery urinary incontinence, including both patient characteristics (such as body mass index, age, prostate volume, comorbidities) and provider-related factors (surgeon experience, skills, center’s volume, etc.) [12,13].

From a pure surgical perspective, several techniques have been recently described in order to achieve early recovery of urinary continence after RALP; among these, the most relevant include bladder neck preservation [14,15], reconstruction of vesicourethral support [16], periurethral suspension [17], total anatomic reconstruction [18], maximal urethral length preservation [19,20,21], endopelvic fascia and/or puboprostatic ligament sparing techniques [13,22,23], combined approaches [24], and pure posterior reconstruction (PR) of the rhabdosphincter [25,26,27,28,29,30].

Although many of these reconstructive techniques aim at restoring the normal anatomical and functional relationships of the pelvic floor, there is no consensus on which reconstruction might be related the most to early continence recovery [31].

Given the absence of absolute recommendations on whether or not to perform these reconstructive procedures, which are time-consuming and might require additional training and experience, in this study we aimed to evaluate the pattern of urinary continence recovery after RALP with no anterior/posterior reconstruction at a referral academic center.

## 2. Materials and Methods

After Institutional Ethical Committee approval, data from consecutive patients undergoing RALP between January 2017 and March 2021 were prospectively collected in our institutional web-based database. Patients were considered eligible for RALP if they had a non-metastatic disease, histologically confirmed prostate cancer, and a life expectancy of more than 10 years. Patients reporting clinically significant urinary incontinence before RALP were excluded from the study.

Diagnosis of prostate cancer was performed with transrectal or transperineal US-guided biopsy, with or without multiparametric magnetic resonance imaging (mpMRI) guidance. All mpMRIs were evaluated by dedicated uroradiologists.

The decision to perform RALP was based on shared decision-making with patients after proper counseling on all potential alternatives (active surveillance or radiation therapy, where indicated). No patient underwent preoperative androgen deprivation therapy. All procedures were performed by three high-volume surgeons (>100 RALP at the beginning of the study period and >40 RALP/year) at our referral institution.

### 2.1. Surgical Technique: Nuances for Urinary Continence Recovery

After the induction of anesthesia, the patients were placed in a modified lithotomy position and a four-arm approach with six trocars was used [32]. The port configuration for RALP is shown in Figure 1A.

RALP was performed following the principles of the Montsouris technique, with a bladder-neck-sparing intent and maximal preservation of the membranous urethra (if oncologically safe) without any anterior/posterior reconstruction. In particular, at the beginning of the procedure, the reflection of the peritoneum in the recto-vesicle pouch (pouch of Douglas) is identified. After identification and dissection of both vas deferens and seminal vesicles, the posterior plane between the prostate and the Denonvillier’s fascia is developed until the prostatic apex. Then, if a neurovascular bundle preservation is planned, an intra- or inter-fascial dissection plane is identified and carefully developed monolaterally or bilaterally (Figure 1B). In case of locally advanced cancer, an extra-fascial (radical) dissection is considered. After the posterior dissection, the bladder is released and dropped down by incising the peritoneum from the right to the left obliterated umbilical artery. The anterior prostatic fat (APF) is dissected to skeletonize the puboprostatic ligaments for optimal visualization of the prostatic apex. The endopelvic fascia is incised bilaterally if a radical intent is pursued. The lateral pelvic fascia and the levator ani fascia are meticulously separated from the prostate following an avascular plane through blunt dissection to avoid any thermal damage.

Then, the plane between the bladder neck and the prostate is carefully developed, ideally with a bladder-neck-sparing intent if oncologically safe (Figure 1C). Upon reaching bladder fibers, the curvature of the prostate in the sagittal plane is followed proximally to the bladder neck. The incision is extended laterally in an arched fashion to avoid vessels that runs from the prostate lateral pedicle to the dorsal vascular complex. Blunt dissection is then performed in a caudal direction over the anterior bladder neck to identify the vertical fibers of the prostatic urethra. Blunt dissection is addressed laterally to the bladder neck on each side, resulting in a triangular spread bilaterally on the lateral lobes of the prostate and defining the funneled shape of the bladder neck transitioning to the prostatic urethra (Figure 1D). An increased urethral length, which includes a greater amount of smooth muscles and the rhabdosphincter, increases the length of the urethra pressure profile (Figure 1E). Once completely dissected, the bladder neck is opened anteriorly, the urethral catheter is withdrawn after deflating the balloon, and the posterior bladder mucosa incised with monopolar energy. This created a foothold to grasp the urethra and elevate the prostate for traction. The dissection then continues laterally to the adipose tissue that defines the lateral border of the prostate-vesical plane. The detrusor apron is opened as low as possible, revealing the previously dissected vas deferens and seminal vesicles. The procedure continues with a nerve-sparing or radical intent according to the individual tumor characteristics. Finally, the DVC is transected to expose the prostatic apex, and then overseen with a 4-0 V-lock suture to prevent delayed venous bleeding. The urethra-vesical anastomosis is performed using the Van Velthoven technique, with two 16 cm 3-0 Monocryl sutures (Figure 1F).

Bilateral extended pelvic lymph node dissection is performed in patients for whom the probability of lymph node metastases exceeded the threshold defined by the Briganti nomogram (5% in the 2012 nomogram for patients who did not undergo preoperative mpMRI; 7% in the 2019 nomogram for those who underwent mpMRI-guided fusion biopsy).

### 2.2. Dataset and Perioperative Protocol

The following data were collected in our prospective database: patient age, body mass index (BMI), diabetes, Charlson comorbidity index (CCI), anticoagulant/antiplatelet medication, cardiovascular disease (defined as a group of disorders of the heart and blood vessels requiring medical treatment), previous abdominal or pelvic surgery, preoperative PSA, clinical stage according to the digital rectal examination (DRE), prostate volume measurement and imaging data. The presence of neurovascular bundle thickening, bulge, loss of capsule integrity, capsular enhancement or measurable extracapsular disease detected at high-volume T2-weighted images in the radiological report was used to define extracapsular extension (ECE) at mpMRI. For grading, ISUP categories, grade group, and the modified Gleason scoring system according to the International Society of Urological Pathology 2005 and 2014 consensus conferences were adopted. Preoperative exams, imaging, and follow-up after surgery were performed following the EAU Guidelines recommendations. Patients were stratified according to either the D’Amico risk groups [33] or the EAU Guidelines risk groups [34].

All patients, during the urethral catheter removal, received detailed information and booklets about pelvic floor muscle training to improve continence recovery after RALP.

### 2.3. Outcome Measure

Urinary incontinence (UI) was recorded according to the Expanded Prostate Cancer Composite (EPIC) Score (EPIC) criteria [35], based on patients’ self-assessment of urinary continence [11]. Specifically, UI was defined as the need of one or more pads per die. UI was recorded at both a short (30 d after RALP) and a mid-term follow-up and stratified as absent, slight (1 pad die), moderate (2 pad/die), or severe (≥3 pad/die) according to the MASTER trial criteria [36].

Oncological outcomes, as well as intraoperative, early (30 d) and late (>30 d) post-operative complications (prospectively according to the modified Clavien–Dindo system [37]) were considered secondary objectives of the study.

### 2.4. Statistical Analysis

Descriptive statistics were obtained reporting median and interquartile ranges (IQR) for continuous variables, while numbers and proportions for categorical variables, as appropriate.

The statistical analysis plan for this study included two steps: first, we obtained the baseline clinical characteristics of patients, focusing on the proportion of those reporting urinary continence recovery. Then, a sub-analysis was performed in patients who underwent preoperative mpMRI. Lastly, we explored the independent predictors of early postoperative incontinence using multivariable logistic regression analysis.

We could not evaluate the predictors of mid-term urinary incontinence due to the small number of events.

Statistical analyses were performed using SPSS v. 26 (IBM SPSS Statistics for Mac, Armonk, NY, USA, IBM Corp). All tests were two-sided with a significance set at *p* < 0.05.

## 3. Results

A total of 925 patients were included. Median age, BMI, and Charlson Comorbidity Index (CCI) were 68 years, 26 kg/m^2^, and 1, respectively. Median PSA value at diagnosis was 6.8 ng/mL (Table 1). In our cohort, 572 patients (61.8%) underwent a nerve-sparing procedure (38.2% monoliteral; 25.6% bilateral). In 487 cases (52.6%), pelvic lymph node dissection was performed (of these, 106/487 [21.7%] harbored pN+ disease). Clinically significant positive surgical margins (>2 mm) were recorded in 7.8% of patients (Table 2). No major intraoperative complications were recorded. Moreover, 18 (1.9%) major early (<30 days) complications were recorded: of these, 15 (1.6%) were CD IIIa, 2 (0.2%) CD IIIb, and 1 CD IV. Similarly, 39 (4.2%) major complications occurred after 30 days (18 CD IIIa, 20 CD IIIb, and 1 CD IV). Biochemical recurrence requiring adjuvant treatment was recorded in 63 cases (6.8%) at a median follow-up of 17 months (range 10–24).

Regarding UI recovery, 82.8 % of patients in our cohort reported to be continent at 1 month after surgery. An improvement in continence recovery was evident during the follow-up, with 94.5% of patients reporting to be fully continent at the last follow-up (Table 3).

In the multivariate analysis, the only independent predictors for early (30 d) continence were the presence of baseline cardiovascular disease and a non-nerve-sparing surgical intent during RALP. None of the other patient- or tumor-related factors, including the EAU risk category, was associated with the risk of UI at a short-term follow-up (Table 4).

Results did not change in the sub-analysis including patients who underwent preoperative mpMRI (Appendix A).

## 4. Discussion

Urinary incontinence after radical prostatectomy may significantly impact patient wellbeing and quality of life [38]. Robotic surgery has revolutionized the surgical technique for radical prostatectomy, leading to the proposal of a variety of technical strategies to improve urinary continence recovery after surgery while ensuring perioperative safety and oncological efficacy [3,39,40,41,42,43]. Continence recovery after RALP is strongly influenced by several intraoperative and post-operative factors and their interplay: despite UI still representing a challenging issue, all surgeons acknowledge that minimizing the risk of UI in both the short- and mid-term postoperative period is one of the main goals of RALP from a functional standpoint.

For this reason, the surgeon should always pursue surgical strategies that aim at preserving the anatomical structures involved in the continence mechanism [44].

In the last few years, different strategies that led to an overall improvement of postoperative functional outcomes have been identified: bladder-neck preservation, reconstruction of puboprostatic ligaments, creation of posterior urethral support, and variation of suspension structures [17,24,25,26,27,44,45,46,47,48,49,50,51,52]. In the most recent illustrative review on this topic, Vis et al. [14] offered insights on how to perform a variety of techniques to improve the chance of continence recovery in the early postoperative period. Although many of the proposed procedures report a benefit with respect to early continence, benefits seem to diminish with longer follow-up. As such, whether any of the reconstructive techniques is superior to another is still matter of debate. Moreover, it should be noted that, to date, only few RCTs have compared a particular reconstructive technique with “no reconstruction” or a different reconstructive technique [5,17,24,26,53,54,55,56,57].

In this scenario, whether specific reconstruction techniques should be routinely employed by surgeons to improve continence recovery after RALP is controversial.

Our study provides several insights for surgeons to contextualize the current debate over the merits and limitations of different reconstruction techniques. First, our findings should be interpreted in light of the specific surgical technique performed by surgeons involved in our series. From a purely technical standpoint, the two main surgical strategies that are always performed are the anatomical dissection of the bladder neck coupled with maximal urethral length preservation. In addition, pubo-prostaic ligaments are routinely spared (even in non-nerve-sparing procedures), while the endopelvic fascia and the pubo-prostatic “Afrodite” veil is entirely spared only in intrafascial nerve-sparing RALPs. Lastly, nerve-sparing procedures are often carried out in a “clipless” fashion to minimize the potential detrimental impact of energy on neurovascular bundles.

Overall, following the principles of the Montsouris technique, our strategy involves careful, anatomical bladder-neck preservation (if oncologically safe) and aims to maximize the length of the membranous urethra during the dissection of the prostatic apex. Yet, it did not involve any specific anterior or posterior reconstruction before/after the urethra-vesical anastomosis, providing an opportunity to analyze the functional outcomes achieved by RALP without reconstructive phases.

The temporal pattern of urinary continence recovery after RALP reported in our study confirms the safety of our technique for RALP (always performed by highly experienced surgeons at a referral institution) from a function viewpoint. In particular, the occurrence of postoperative UI at a mid-term follow-up (median follow-up of 17 months) was a rare event (<5%). On the contrary, we noted a non-negligible proportion of patients (self)-reporting UI at a short-term follow-up (30d), suggesting that the very immediate postoperative period offers further chances of improvement. In the multivariable analysis adjusting for several patient-, tumor- and surgery-related features, only the baseline presence of cardiovascular diseases and the performance of RALP with a radical (non-nerve-sparing) intent were associated with a higher risk of experiencing short-term UI. Taken together, our data suggest that patient-related characteristics (i.e., a higher comorbidity burden), rather than specific technical nuances, might decrease the chance of full continence recovery after RALP in our cohort. Yet, in our analysis we could not analyze the differential impact of specific steps of the procedure, such as bladder-neck preservation, nerve-sparing technique, etc., on the risk of early UI, mainly due to lack of granular data on these technical details for all patients.

Importantly, our sub-analysis including only patients who underwent a mpMRI-guided diagnostic pathway for prostate cancer diagnosis (namely preoperative mpMRI followed by mpMRI-guided fusion biopsy) confirmed the findings reported for the overall cohort, suggesting that the information gained by surgeons from mpMRI (which might have guided the performance of a nerve-sparing procedure and other strategic steps of RALP) did not significantly impact on the risk of UI at both a short- and mid-term follow-up. Lastly, mirroring the results in the overall cohort, in a multivariable model including only patients who underwent preoperative mpMRI, neither established tumor-related features nor the four novel categories predicting the risk of early biochemical recurrence proposed by Mazzone et al. [34] based on clinical and radiological parameters were found to be significant predictors of early UI.

Our findings, which come from a tertiary referral academic center and which are grounded on an established surgical technique employed by high-volume surgeons, provide a robust foundation for implementing preoperative patient counseling regarding the expected functional outcomes of RALP in daily clinical practice. In particular, while patients who are candidates for RALP should be reassured that, in experienced hands, postoperative UI is a rare “adverse event” in the mid-term follow-up, at the same time they should be properly informed regarding the risk of transient stress UI in the short-term period, occurring in almost one out of five patients in our series. In this regard, the implementation of specific anterior/posterior reconstructive techniques, as proposed by several authors [14], might improve the chance of early continence recovery and should be the object of high-quality prospective (ideally randomized) clinical trials involving surgeons of different skills, backgrounds, and volume.

Our study is not devoid of limitations. First, despite prospective data collection in the pre- and perioperative period, the assessment of UI during the follow-up is prone to attrition bias and detection bias. Second, the evaluation of UI after RALP was based on patients’ self-assessments rather than objective metrics, which could have influenced our results. Third, we could not evaluate the proportion of patients experiencing UI at longer follow-up periods, limiting the generalizability of our findings. Third, our cohort included patients who underwent RALP by highly experienced surgeons; as such, our UI rates could not be directly interpreted in our clinical scenarios. Lastly, we could not evaluate the differential impact of specific technical steps of RALP on the risk of UI. We also have missing data about which were the specific modalities and pelvic floor muscle training that allowed 95% of the patients to achieve complete urinary continence at their last follow-up.

## 5. Conclusions

In experienced hands, most patients fully recover urinary continence after RALP performed without specific anterior or posterior reconstruction techniques at a mid-term follow-up. On the contrary, a non-negligible proportion of patients reported stress urinary incontinence 1 month after surgery, suggesting that the implementation of surgical reconstructive techniques might still improve continence recovery in the early postoperative period. Prospective multicenter studies are needed to confirm the need of and to select the best candidates for anterior and/or posterior reconstructive techniques after RALP.

## Figures and Tables

**Figure 1 jcm-12-01358-f001:**
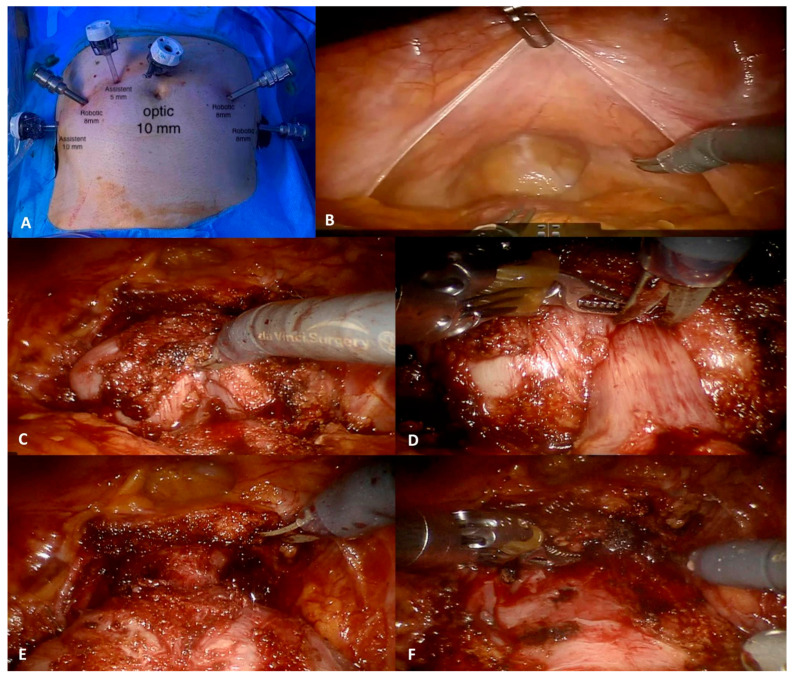
The University of Florence Technique for bladder-neck-sparing robot-assisted radical prostatectomy, step-by-step procedure. (**A**) Trocar mapping; (**B**) incision at the level of refection of peritoneal in recto-vesicle pouch (pouch of Douglas); (**C**) dissection of the vescico-prostatic margin; (**D**) Defining the funneled shape of the bladder neck transitioning to the prostatic urethra; (**E**) Isolation the urethra while preserving the maximum possible length; (**F**) urethrovesical anastomosis was performed using the Van Velthoven technique.

**Table 1 jcm-12-01358-t001:** Baseline patient and tumor characteristics in the overall cohort.

Variables	*n* = 925
Patient’s characteristics	
Age at diagnosis, yrs, median (IQR)	68 (63–72)
BMI at surgery, kg/m^2^, median (IQR)	26 (24.0–28.0)
Charlson Comorbidity Index (CCI)	1 (0–2)
Cardiovascular disease *n* (%)	244 (26.4)
Diabetes mellitus, *n* (%)	80 (8.6)
Anticoagulant/antiaggregant, *n* (%)	226 (24.4)
Family history of prostate cancer, *n* (%)	80 (8.6)
Preoperative urinary incontinence (UI) *, *n* (%)	145 (15.7)
Preoperative ED, *n* (%)	145(15.7)
Positive DRE, *n* (%)	473 (51.1)
PSA value at diagnosis, ng/mL, median (IQR)	6.8 (4.7–9.7)
Preoperative MRI, *n* (%)	518 (56)
Prostate volume, cc, median ° (IQR)	45 (34.0–54.0)
Characteristics at biopsy	
Type of biopsy, *n* (%)	
Systematic biopsy	520 (56.2)
Cognitive biopsy	51 (5.5)
MRI-targeted biopsy	349 (37.7)
Incidental prostate cancer following TURP.	5 (0.5)
Highest ISUP Grade Group at random ^ç^ biopsy cores, *n* (%)	
Patients with no positive random biopsy cores	114 (12.3)
Grade Group 1	187 (20.2)
Grade Group 2	284 (30.7)
Grade Group 3	183 (19.8)
Grade Group 4	102 (11)
Grade Group 5	55 (5.9)
Extra-capsular invasion at biopsy, *n* (%)	16 (1.7)
Seminal vesicle invasion at biopsy, *n* (%)	8 (0.9)
Perineural invasion at biopsy (PNI), *n* (%)	237 (25.6)
Characteristics at imaging	
Preoperative staging	751 (81.1)
Extra-capsular extension at imaging, *n* (%)	50 (6.5)
Seminal vesicle invasion at imaging, *n* (%)	23 (3.0)
cT, *n* (%)	
T1	424 (45.8)
T2	467 (50.5)
T3–4	34 (3.7)
cN+, *n* (%)	64 (6.9)
Localized disease ^§^	839 (90.7)
EAU class risk stratification, *n* (%)	
Low-risk disease	117 (12.6)
Intermediate-risk disease	438 (47.3)
High-risk disease	289 (31.2)
Locally advanced disease	81 (8.7)

* UI defined according to EPIC criteria as the need for of more than one Pad per day; ° Prostate volume was assessed at sovrapubic ultrasound and/or MRI, if available; ^ç^ The grade group was evaluated only on patients with random samples at biopsy (*n* = 925). If case of multiple Gleason scores in the same patient, the highest was considered; ^§^ Localized disease was defined as the presence of cT1-2, N0 M0 prostatic cancer, according to EAU Guidelines. BMI: body mass index; ED: erectile dysfunction; MRI: magnetic resonance imaging; TURP: transurethral resection of prostate.

**Table 2 jcm-12-01358-t002:** Intra and histopathological features for the overall cohort.

Variables	*n* = 925
Surgical features	
Nerve-sparing RALP, *n* (%)	572 (61.8)
Monolateral nerve-sparing RALP *n* (%)	353 (38.2)
Bilateral nerve-sparing RALP *n* (%)	237 (25.6)
Pelvic lymph node dissection during (LND) RALP, *n* (%)	487 (52.6)
Patient with locally advanced disease undergoing nerve-sparing RALP, *n* (%)	24 (2.6)
Patients with high-risk or locally advanced EAU disease underging pelvic lymph node dissection, *n* (%)	299 (32.3)
Histopathologic features	
Definitive histology findings, *n* (%)	
Acinar adenocarcinoma	876 (94.7)
Intraductal carcinoma	5 (0.5)
Mixed	14 (1.5)
Others (sarcomatoid, squamous and adenosquamous)	30 (3.2)
Clinically significant positive surgical margin (PSM), more than 2 mm, *n* (%)	72 (7.8)
Highest ISUP grade group post-RP histopathologic assessment, *n* (%) °	
Grade Group 1	81 (8.9)
Grade Group 2	400 (43.8)
Grade Group 3	182 (19.9)
Grade Group 4	154 (16.8)
Grade Group 5	97 (10.6)
Gleason Score (GS) at definitive post-RP histopathologic assessment, *n* (%)	
GS 6 (3 + 3)	81 (8.9)
GS 7 (3 + 4 and 4 + 3)	619 (67.7)
GS 8 (4 + 4 and 5 + 3 and 3 + 5)	129 (14.1)
GS 9 (4 + 5 and 5 + 4)	85 (9.3)
Perineural invasion (IPN) at at definitive post-RP histopathologic assessment, *n* (%)	428 (46.3)
pT, *n* (%)	
T2	349 (37.8)
T3	574(62.0)
T4	2 (0.2)
pN, *n* (%)	
Nx	438 (47.4)
N0	381 (41.2)
N1	106 (11.5)
pM, *n* (%) °°	
Mx	882 (95.4)
M0	40 (4.3)
M1 (extraregional LND)	3 (0.3)

° Calculated for patients with acinar adenocarcinoma at definitive histology or with acinar-intraductal mixed tumors with acinar prevalence (*n* = 914). °° These 3 patients are those who have performed a super-extended lymphectomy with evidence of non-regional lymph node metastases (stage M1a).

**Table 3 jcm-12-01358-t003:** Post-operative outcomes and UI outcomes at both short- and mid-term follow-up.

Variables	*n* = 925
Follow-up	
Duration of f-up, months, median (IQR)	17 (11-27)
Biochemical recurrence (BCR)at last follow-up	63 (6.8)
-Of patients with BCR:	
-adjuvant radiotherapy (in 10 cases concomitant ormonotherapy)	27 (42)
-salvage radiotherapy	8 (12)
-systemic ormonotherapy	9 (14)
Early (30 day) continence rate, *n* (%) *	
Fully continent, no pad/die	766 (82.8)
1 or more pads/dies	159 (17.2)
Patients with moderate/severe incontinence within 30 days	
2 pads/dies	125 (78.6)
3 pads/dies	26 (16.4)
>3 pads/dies	8 (5)
Continence at last follow-up days, *n* (%)	
Fully continent, no pad/die	885 (95.5)
1 or more pads/dies	40 (5.5)
Patients with moderate/severe incontinence after 30 days	
2 pads/dies	25 (62.5)
3 pads/dies	14 (35)
>3 pads/dies	1 (2.5)
Clavien–Dindo (CD) > 2 (within 30 days), *n* (%)	
CD IIIa *(10 cases requiring percutaneous drainage for symptomatic lymphocele and 5 requiring endoscopic catheter positioning for mdc spread)*	15 (1.6)
CD IIIb *(1 case clot retention requiring endoscopic evacuation and 1 case requiring re-intervention for uretrovesical anastomosis)*	2 (0.2)
CD IV *(1 case of post-operative cerebrovascular ischemia requiring neurovascular thrombolysis)*	1 (0.1)
Claviend Dindo (CD) > 2 after 30 days), *n* (%)	
CD IIIa *(13 cases requiring percutaneous drainage for symptomatic lymphocele, 5 percutaneous drainages for abdominal abscessus)*	18 (1.9)
CD IIIb *(13 endoscopic urethrotomy for endoscopic stricture, 4 paraumbilical haernia requiring surgical repair, 1 re-operation for bladder repair, 1 case requiring re-intervention for uretrovesical anastomosis and 1 case requiring endoscopic surgery for clip removal)*	20 (2.1)
CD IV *(1 case of acute abdomen for bowel strangulation)*	1 (0.1)

* Evaluated on patients for whom it is possible to recover data on continence (*n* = 873) also by means of a remote interview.

**Table 4 jcm-12-01358-t004:** Multivariable logistic regression analysis assessing the predictors of short-term UI in our cohort.

Multivariate Logistic Regressions for all Patients (*n* = 925)				
Variables	OR	*p*	95% Confidence Interval	
			Inferior	Superior
Cardiovascular disease (absent vs. present at diagnosis)	0.463	<0.01	0.321	0.667
Pelvic lymph node dissection during RALP	1.337	0.187	0.868	2.058
Nerve-sparing (vs. radical) RALP	1.522	0.045	1.007	2.300
EAU low-risk disease referent	-	-	-	-
EAU intermediate-risk disease	0.967	0.941	0.401	2.330
EAU high-risk disease	1.183	0.632	0.596	2.347
EAU locally advanced disease	1.189	0.618	0.602	2.347

RALP: Robot-Assisted Laparoscopic Prostatectomy; EAU: European Association of Urology.

## Data Availability

The data presented in this study are available on request from the corresponding author.

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
