# Peer review of "Urinary Continence Recovery after Robotic Radical Prostatectomy without Anterior or Posterior Reconstruction: Experience from a Tertiary Referral Center"

_jcm, 2023, doi:10.3390/jcm12041358_

Round 1

Reviewer 1 Report

This study analyzed the results of urinary continence recovery for the authors' surgical techniques . The manuscript is well written. However, minor issue should be addressed.

1.The title of the paper is too broad and the purpose of the study mentioned in the ‘Abstract’ is also comprehensive.

This study was intended to report the continence recovery of the authors' specific surgical technique (RALP without reconstruction), so it is recommended that the title and purpose of the abstract be modified accordingly.

2.The sentence below in the result part corresponds to discussion.

‘Unlike what recently reported in the current literature [11], androgen depriving therapy (ADT) wasn’t associated to UI at long term follow-up: this can probably be explained by the young age of this group of patients.’

3. Table 3 is misspelled as Table 1.

Table 5 is misspelled as Table 4.

Author Response

Dear reviewer, thanks for your comments. We are glad to give the explanations for your concerns:

  1. The title of the paper is too broad and the purpose of the study mentioned in the ‘Abstract’ is also comprehensive. This study was intended to report the continence recovery of the authors' specific surgical technique (RALP without reconstruction), so it is recommended that the title and purpose of the abstract be modified accordingly.

We thank the reviewer for this comment. We edited the title and the abstract according to the reviewer suggestion.

  1. The sentence below in the result part corresponds to discussion. “Unlike what recently reported in the current literature [11], androgen depriving therapy (ADT) wasn’t associated to UI at long term follow-up: this can probably be explained by the young age of this group of patients”.

This sentence was meant to be in the discussion, we are sorry for not taking it into account. However, since this sentence refers to a topic that has been already discussed in that section, we decided to delete it rather than moving it.

  1. Table 3 is misspelled as Table 1.Table 5 is misspelled as Table 4.

The table are now spelled correctly.

Reviewer 2 Report

COMMENTS TO  JCM-2169498

Urinary incontinence after radical prostatectomy is a major problem. Therefore, the article deals with a topic of maximum interest.

The article is well structured, but it provides incredible continence figures after radical prostatectomy: that is, a continence rate never reported before.

It requires the following clarifications:

1.-Define what you call a very experienced urologist. Does this mean that the study presented has been carried out on patients operated on by surgeons who had previously operated on more than 30 robot-assisted prostatectomies?

2.-The authors do express the weakness that the results are measured by what the patients communicate, without an objective measure of urinary continence. This may explain a continence rate higher than that of any experienced urological team in hospitals with a high level of care.

3.- It is important that the relationship between the possibility of preservation of neurovascular bundles, bladder neck, and tumor stage be clearly clarified. Given that the preservation of neurovascular bundles and bladder neck must be related to a tumor stage that allows tumor removal with oncological safety.

Author Response

Dear reviewer, thanks for your comments. We are glad to give the explanations for your concerns:

  1. Define what you call a very experienced urologist. Does this mean that the study presented has been carried out on patients operated on by surgeons who had previously operated on more than 30 robot-assisted prostatectomies?

All the surgeries were conducted by surgeons that performed more than 100 RALP at the beginning of the study period  and more than 40 RALP per year. You can now find in the text our definition of experienced surgeon, accordingly .   

  1. The authors do express the weakness that the results are measured by what the patients communicate, without an objective measure of urinary continence. This may explain a continence rate higher than that of any experienced urological team in hospitals with a high level of care.

We thank the Reviewer very much for this comment. We indeed used a self-assessment method to report UI recovery after RALP. However, we believe that, ultimately, patient-reported outcomes are the most important outcomes in this field and should be central in shared decision-making. While more objective measurements could have been employed, these might have been biased by clinicians at the time of the evaluation. We have specified that our results might have been influenced by this choice in the limitations. Moreover, almost one out of five patients did NOT recover UI at 1 month (this is worse than reported in other series by experienced Centres), while they recovered it almost entirely during the follow-up (this is coherent with available series). Lastly, the incidence of UI after RALP varies with the definition used (https://doi.org/10.3389/fsurg.2021.647656)

  1. It I s important that the relationship between the possibility of preservation of neurovascular bundles, bladder neck, and tumor stage be clearly clarified. Given that the preservation of neurovascular bundles and bladder neck must be related to a tumor stage that allows tumor removal with oncological safety.

We entirely agree with the Reviewer. In fact, we have included these concepts in the multivariable analysis (Table 3), analyzing as predictors the EAU risk categories (based on tumor stage, PSA and DRE), the performance of LND (surrogate metrics of higher aggressiveness) and nerve-sparing procedures (surrogate metrics of tumor location within the prostate as well as tumor stage).